# Challenges When Evaluating Cognitive Bias Modification Interventions for Substance Use Disorder

**DOI:** 10.3390/ijerph17217821

**Published:** 2020-10-26

**Authors:** Melvyn W. B. Zhang, Helen E. Smith

**Affiliations:** 1National Addiction Management Service, Institute of Mental Health, Singapore 539747, Singapore; 2Family Medicine and Primary Care, Lee Kong Chian School of Medicine, Nanyang Technological University Singapore, Singapore 308232, Singapore; h.e.smith@ntu.edu.sg

**Keywords:** attention bias, cognitive bias, Psychiatry

## Abstract

In recent years, advances in experimental psychology have led to a better understanding in automatic, unconscious processes, referred to as attentional and approach biases amongst individuals with substance use disorders. Attentional biases refer to the relatively automatic tendencies for attention to be preferentially allocated towards substance-related cues. Whereas, approach bias refers to the relatively automatic behavioral tendencies of individuals to reach out to substance-related cues in their natural environment. While, several reviews confirm the existence of these biases, and the effectiveness of bias modification, the conduct of cognitive bias modification amongst substance-using individuals is not without its challenges. One of these is that cognitive biases, both attentional and approach biases, are not universally present; and several individual differences factors modulate the magnitude of the biases. Another challenge that investigators faced in their conduct of cognitive bias modification relates to the selection of the appropriate task for bias assessment and modification. Other challenges intrinsic to cognitive bias modification intervention relates to that of participant attrition, much like conventional psychotherapies. Negative findings, of the absence of biases at baseline, or the lack of effectiveness of bias modification have been reported in studies of cognitive bias modification. All these challenges could have an impact on bias assessment and modification. In this perspective paper, we will explore the literature surrounding each of these challenges and discuss potential measures that could be undertaken to mitigate these clinical and research challenges.

## 1. Introduction

In recent years, the advances in experimental psychology have led to there being better understanding of relatively automatic processes, referred to as attentional and approach biases [1]. For substance use disorders, attentional biases refer to the relatively automatic tendencies for attention to be preferentially allocated towards substance-related cues [1,2]. Approach biases are the relatively automatic behavioral tendencies of individuals to reach out to substance-related cues in their natural environment [2,3]. This attraction is a predisposing factor, leading individuals to relapse into their underlying addictive disorders. The advances in experimental psychology have demonstrated that conventional psychotherapies do not impact on these more reflexive and relatively automatic processes; this is because conventional psychotherapies target conscious, goal directed cognitive control processes. There are several reviews that have provided evidence for the existence of biases. For example, MacLean RR et al. (2018) [4], in their review, describe the evidence for the existence of robust attentional biases amongst individuals who were using opioids. Similarly, O’Neil et al. (2020) [5] reported that attentional biases are greater in individuals abusing cannabis. Two other reviews have focused on the effectiveness of bias modification. Boffo M et al. (2019)’s [6] most recent review has advanced the work undertaken by Cristea et al. (2016) [2], by undertaking a Bayesian network meta-analysis, and the authors reported that the effect size for cognitive bias modification to be 0.23, amongst a sample of individuals with alcohol and tobacco disorders. 

While, these reviews confirm the existence of biases, and the effectiveness of bias modification, the conduct of cognitive bias modification amongst substance-using individuals is not without its challenges. One of these is that the magnitude of attentional and approach biases is dependent on individual factors. MacLean RR et al. (2018) [4] reported the existence of robust biases amongst opioid users in their meta-analysis, but other individual studies have highlighted a dose-dependent relationship between the amount of opioids used and the magnitude of attentional biases (Bearre L et al., 2007) [7]. This relationship is not limited to opioid use disorders; Field M et al. (2005) [8] similarly reported that attentional biases amongst cannabis users were dependent on both, the frequency of cannabis use and their subjective craving scores. Such individual differences (severity of dependence, frequency of drug use) in baseline attentional biases will impact on the effectiveness of cognitive bias modification interventions. It is apparent that cognitive bias modification appears to work best for individuals who have clinical levels of dependence, or whom treatment seeking (Wiers et al., 2018) [8]. In fact, Dennis-Tiwary et al. (2016) [9] were amongst the first to highlight how individuals differ; and that these individual differences could potentially have an impact on the effectiveness of a bias modification intervention. They highlighted that an understanding of individual differences would enable personalization of the cognitive bias modification intervention, resulting in greater modification of attentional biases.

Another challenge that investigators faced in their conduct of cognitive bias modification relates to the selection of the appropriate task for bias assessment and modification. As Zhang et al. (2018) [10] highlighted, there are both direct (such as measurement of eye fixation time) and indirect measures (Stroop and Visual Probe task) of assessing and modifying biases and each have their weaknesses. The visual probe task, which has been commonly used, involves having the individual respond to the position of a probe replacing either the substance or neutral stimuli. Individuals with biases tend to respond faster to probes that replace the substance stimuli, as compared to the neutral stimulus. There are reliability issues with indirect tasks, coupled with the fact that there remains huge variability in the nature of the paradigms used (Zhang et al. (2019)) [11] and the stimulus timing intervals has an impact on the assessment of attentional biases. Presenting stimulus images at relatively fast stimulus intervals, such as those less than 250 milliseconds, is more likely to capture the reflexive attentional orienting; as compared to presentation at longer stimulus intervals, whereby participants chose where to look on the screen [12]. This is evident, as Garland et al. (2013) [13], have reported that in opioid users a stimulus timing interval of 200 milliseconds is better capturing biases than a 2000 millisecond interval. 

Other challenges intrinsic to cognitive bias modification intervention relates to that of participant attrition, much like conventional psychotherapies. The large number of repetitions involved in the intervention can diminish motivation over time. To overcome this gamification technologies have been considered, to help maintain participants motivation to train. Zhang et al. (2018) [14] in their review of gamified cognitive bias modification intervention reported mixed findings for the use of gamification. One remaining challenge relates to the development of cognitive bias modification interventions. When Zhang et al. (2018) [15] reviewed mobile applications in the literature and in commercial stores, they highlighted a disconnect between academics and developers. This is concerning, as it implies that there are applications without evidence base; and scientifically evaluated applications that are not on commercial stores. 

One last challenge that researchers face with regards to bias modification interventions pertains to that of negative findings. Negative findings could refer to either the absence of baseline biases amongst individuals during the assessment stage; or the absence of positive findings following bias modification. Our previous study, which was a feasibility and acceptability study examining a mobile attention bias modification intervention revealed that 53% of the participants sampled to have no baseline biases [16]. Dean AC et al. (2019) [17] in their recently concluded study examining attentional bias modification amongst 42 methamphetamine-dependent individuals reported negative findings following intervention, i.e., bias retraining did not reduce attentional biases. Zhu Y et al. (2018) [18] in a study that examined a computerized cognitive addiction therapy application for individuals with methamphetamine disorders also reported that the cognitive bias retraining functionality in the app did not reduce overall biases. For approach/avoidance biases, Barkby et al. (2012) [19] have previously examined 63 alcohol-dependent participants undergoing detoxification against 64 controls, who were light drinkers. They found, based on their stimulus-compatibility task, that there were no differences between dependent drinker and light drinkers on the task. The only exception was that those who drank more had a greater magnitude of approach biases. Spruyt A et al. (2012) [20] also examined for approach/avoidance biases amongst individuals who were alcohol dependent and amongst controls. The authors reported that their sampled participants demonstrated avoidance tendencies to stimulus, instead of the typical approach tendencies. There have also been other studies have highlighted that the magnitude of baseline biases, in elderly patients, predicted whether bias modification was effective (Eberl C et al., 2012) [21]. There warrants further understanding of why there are negative findings, and how baseline biases correlate with eventual outcomes of bias retraining. 

It is clear from the literature that there are several variables that modulate whether biases are present at baseline, and whether bias modification would be effective. The aim of this perspective paper is really in highlighting evidence for each of these identified challenges and discuss potential measures that could be undertaken to mitigate against them. This perspective article thus helps to map and scope out each of these challenges that future researchers ought to be mindful of. 

## 2. Challenges Pertaining to Individual Differences in Cognitive Biases

### 2.1. A Brief Review of the Literature Was Conducted in Order to Identify Some of the Relevant Articles

#### 2.1.1. Opioid Use Disorders

Both the severity of the dependence and the amount of substance used, is associated with the presence of bias as well as their magnitude. This is demonstrated in Fadardi JS et al. (2010) [22]’s work, whereby they examined a cohort of 53 Iranian drug abusers on methadone maintenance therapy and 71 non-abusers. They reported that drug users had higher biases for drug-related stimulus, even after controlling for age and education. Similar findings were reported in the studies of Constantinou et al. (2010) [23] and Bearre L e al. (2007) [7]. Constantinou et al. (2010) [18] examined attentional biases and cravings amongst current users of opioids, ex-users of opioids and non-users. They found that those who were currently abusing had a greater magnitude of attentional biases, and that ex-users exhibited a bias away from the drug-related stimulus, which was related to their total duration of abstinence. Bearre L et al. (2007) [7] study reported a relationship between dependence severity and the overall attentional biases. 

Other factors that appeared to modulate cognitive biases amongst opioid using individuals include that of impulsiveness, cravings, temptations to use, as highlighted by Anderson BA et al (2013) [24] and Water AJ et al. (2012) [25].

#### 2.1.2. Cannabis Use Disorders

Like Opioid Use Disorders, there is also evidence that both the severity of dependence and the quantity of cannabis consumed modulates the magnitude of the underlying attentional biases. In their meta-analysis, O’Neill A et al. (2020) [5] highlighted that biases were more evident amongst cannabis users. Other studies, such as that of Field M et al. (2006) [26] have reported that it is not just the frequency of use, but also the number of joints smoked that affect the absolute magnitude of attentional biases. Two studies by Cousijin, (Cousijin J et al. (2011) [27] and Cousijin J et al. (2013)) [28], report attentional biases mainly amongst heavy users. In Campbell DW et al. (2018) [29] study comparing users and non-users’ attentional biases, they reported that the intensity of cannabis used (grams used per week), affected the magnitude of attentional biases. Additionally, factors like subjective cravings and perceived stress appear to affect the magnitude of these biases (Field M et al. (2005) [30] and Vujanovic AA et al. (2016)) [31].

#### 2.1.3. Stimulant Use Disorders

In stimulant use disorders, there appears to be a correlation between the severity of the substance dependence and attentional biases, like that observed for opioid and cannabis use disorders. Marks KR et al. (2014) [32] measured the fixation time for the computation of attentional biases and reported that only individuals who have had previously used cocaine had underlying attentional biases. They also reported that the magnitude of the attentional biases was correlated with the lifetime use of cocaine. 

From the published literature, several factors modulate the magnitude of baseline biases. Across opioid, cannabis and stimulant use disorders, the most frequent factors identified are severity of the dependence and the quantity of substance used.

## 3. Challenges Pertaining to the Nature of the Intervention Task

One of the other challenges faced by researchers investigating cognitive bias modification relates to the task paradigm used for the assessment and modification of biases. As aforementioned in the introduction, there are both direct and indirect measures that could be used for assessment. Indirect measures have the advantage of being less invasive as compared to direct measures but are less reliable. There have been recent attempts to improve the reliability of the visual probe task by personalizing the stimulus used, but this has not resulted in as much improvement as expected. 

Zhang et al. (2019) [11], in their review of the visual probe paradigms in the published literature, found many variations in paradigms. This not only affects the reproducibility, but also affects the conceptualization of new interventions. Across the different substance disorders, there appear to be differences in the task parameters that render interventions to be successful. For example, Garland (2013) [11] investigated whether variation in the stimulus timings for the visual probe task impacted on the magnitude of attentional biases and reported that attentional biases were present only when stimulus were presented for short intervals of 200 milliseconds, instead of 2000 milliseconds. However, for Cannabis Use Disorders, Vujanovic AA et al. (2016) [31] reported larger attentional biases when patients were presented with stimulus for short timings, such as 125 milliseconds. This finding has also been reported in O’Neill A et al. (2020) [5]’s meta-analysis, in which they reported larger effects when the stimulus was presented for short timing intervals of 125 to 500 milliseconds. It is known that presentation at short and long stimulus intervals assess for different aspects of cognitive biases. By presenting for short intervals, it assesses initial orientation, whereas the longer presentation assesses for delayed disengagement. There remains much uncertainty and a lack of consensus with regards to which task parameters would render the intervention to be more effective. 

## 4. Challenges Pertaining to Attrition and Motivation to Train

One of the inherent challenges with cognitive bias modification intervention is the repetitiveness of the intervention and resultant boredom; participants are required to complete multiple repetitions of the same task, to affect a shift in their attentional or approach biases. Recently there has been investigations using serious games (games designed specifically for specific intervention) and gamification technologies to reduce the boredom associated with these interventions in the hope of reducing attrition. Zhang et al. (2018) [14] in their review of gamified cognitive bias modification interventions, reported only two out of four interventions to be effective, and these two were for individuals with anxiety disorders. One of the studies, (Boendmarker et al. (2016)) [33] reported there being no added effectiveness with the integration of gamification elements; despite their previous article suggesting various methods that gamification could be added onto a conventional task. As well as gamification other approaches have been considered, including the integration of motivational support as an adjunct to conventional bias modification tasks. This proposal is still under evaluation and hence no definitive evidence available. 

## 5. Challenges Pertaining to Evidence Base of Cognitive Bias Modification Interventions

While, Zhang et al. (2018) [14] have in their review of mobile cognitive bias modification interventions highlighted the presence of a disconnect between academic and developers, it must be noted that most of the existing cognitive bias modification interventions have been conceptualized based on inputs from academics. There remains a lack of patient participation in the conceptualization process, so the intervention might not meet their needs or preferences. Zhang et al. (2019)’s [34] study was perhaps the first study to have explored patient participants’ perspectives on conventional attention bias modification interventions. Patients’ views subsequently contributed to an intervention codesigned with researchers and health professionals. 

## 6. Discussion

Researchers intending to embark on cognitive bias interventions need to be aware that their intervention might yield negative or null results. These challenges of research on this field are not insurmountable and below we consider some of the approaches that could be adopted. Once cognizant of the factors modulating attentional biases researchers can use these to inform the inclusion and exclusion criteria for studies attempting to determine the effectiveness of bias modification interventions. The literature also calls for a more detailed understanding of substance use prior to patients’ enrolment into trials. Tools, such as the addiction severity index can help researchers capture details of substances participants have abused, and the quantity and frequency of substances use within the recent month, or in the past one year. In studies published to date we have not observed consistency in assessing for baseline biases, before participants are enrolled into research involving bias modification.

As highlighted previously, there remain to be huge variations in the visual probe paradigms, which would affect not only the conceptualization of new interventions but also the reproducibility of previous research findings. We urge researchers to be more transparent in their reporting and to share in full the details of the task used, including all the timings. Zhang et al. (2019)’s [11] prior review has highlighted some of the commonalities in the timings used for interventions involving participants with opioid use and cannabis use disorders, which future research could consider. There remains a need to explore if stimulus should be presented at both short or long stimulus interventions, to better measure both initial orientation and delayed disengagement; or for stimulus to be only presented at one interval. 

A further challenge for researchers has been attrition and poor motivation to train. Research into the use of gamification strategies or serious games elements for cognitive bias modification interventions are still in their infancy, Zhang et al. (2018)’s [13] found only four prior studies exploring the use of such elements. Further research in needed in this context, recognising that gaming elements have already been applied successfully to a variety of interventions, including chronic disease rehabilitation, promoting physical activity levels, and supporting mental health. Additionally, Lau et al. (2016) [35] have reported that serious games can improve psychiatric symptoms, with an effect size of 0.55. The gamified approach might help to address, not only the motivation to train and reduce attrition, but also to enhance the overall effectiveness of the intervention. In addition, a codesign approach ought to be considered for future cognitive bias modification interventions, as opined by Zhang et al. (2019) [34]. The adoption of participatory research methods, bringing together the opinions of all key stakeholders, ensures that the application is evidence-based, whilst also acceptable and useful for patient-participants. Co-design address not only the academic developer disconnects, but the disconnect with patients.

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
