# Peer review of "Challenges When Evaluating Cognitive Bias Modification Interventions for Substance Use Disorder"

_ijerph, 2020, doi:10.3390/ijerph17217821_

Round 1

Reviewer 1 Report

In this brief review the authors highlight several important issues that researchers need to be aware of when planning attentional bias modification studies. I think that the paper is well written and my only major comment would be that most of this information is not novel, it having been summarised previously by other authors. However I will leave that up to the editor's discretion.

Specific comments

  1. I don't think it does the field any favours to continually refer to approach and attentional biases as being "automatic" and "unconscious". I don't think anyone has actually provided any evidence to suggest that they are uncontrollable and unconscious. I would encourage the authors to use the phrase "relatively automatic" and remove the word 'unconscious' from the manuscript. I think the nice point made at line 31 still stands but perhaps the authors could say "conventional therapies to do not impact these more reflexive and relatively automatic processes; this is because conventional therapies target goal-directed control processes" rather than using the unconscious/conscious dichotomy.
  2. Line 58 (and 168-175; 214-218): relevant to the issue of timings of attentional bias - the authors could point out that very fast stimulus intervals (i.e. <250ms) the paradigm is more likely to capture fast/reflexive attentional orienting to signals of drug reward but that at longer stimulus intervals participants are strategically choosing where to look on screen (Godijn & Theeuwes, 2002). In the latter case this creates a lot of noise in the data and 'relatively automatic' attentional biases are no longer being measured.
  3. Somewhere (maybe at line 70) I think the authors should point out that cognitive bias modification seems to work best for individuals who have clinical levels of dependence and are treatment seeking, rather than those who wish to merely "cut down" (Wiers et al., 2018).
  4. At line 72 the authors refer to a study on chronic pain in children to make the point that sometime there is no baseline bias. I think this reference seems out of place. There are plenty of studies demonstrating that there is often a lack of baseline bias in clinically dependent populations e.g.,Barkby et al., 2012; Spruyt et al., 2012) which would seem more relevant here.
  5. Related to baseline biases the authors should point out somewhere that Eberl and colleagues reported that "older patients and patients with a strong approach-bias at pretest profited most from CBM" (Eberl et al., 2013)
  6. I find line 75 a bit confusing "the preparation of this perspective paper" - I was thinking - "which paper - The Sharpe one?" To avoid confusion the authors could say "This issue of baseline biases prompted the current perceptive paper after a recent feasibility and acceptability study reported.." I would make this a separate paragraph at the end of the introduction, clearly explaining what the current perspective paper is about and defining the scope.
  7. Similarly when the authors start discussing the search of the published literature at line 84 it is unclear if they are talking about their own search or Dennis-Tiwary et al's search. This information about individual differences should surely go earlier in the introduction, before the authors start discussing their methods? I think the authors should consider have a brief 'method' section where they describe the search they did using Pubmed and MEDLINE - how many papers were returned? How did the authors decide which papers to include? If the authors do not wish to follow guidelines on systematic review then maybe it is easier to just omit the search methodology altogether (and not include a methods section). But i still think the scope of this review needs to be defined somewhere.
  8. Line 79: I think somewhere in the introduction, when 'individual differences' are discussed the authors should give examples of what types of factors they mean.
  9.  

References

Barkby, H., Dickson, J. M., Roper, L., & Field, M. (2012). To Approach or Avoid Alcohol? Automatic and Self-Reported Motivational Tendencies in Alcohol Dependence. Alcoholism: Clinical and Experimental Research, 36(2), 361–368. https://doi.org/10.1111/j.1530-0277.2011.01620.x Eberl, C., Wiers, R. W., Pawelczack, S., Rinck, M., Becker, E. S., & Lindenmeyer, J. (2013). Approach bias modification in alcohol dependence: Do clinical effects replicate and for whom does it work best? Developmental Cognitive Neuroscience, 4, 38–51. https://doi.org/10.1016/j.dcn.2012.11.002 Godijn, R., & Theeuwes, J. (2002). Programming of endogenous and exogenous saccades: Evidence for a competitive integration model. Journal of Experimental Psychology. Human Perception and Performance, 28(5), 1039–1054. Spruyt, A., De Houwer, J., Tibboel, H., Verschuere, B., Crombez, G., Verbanck, P., Hanak, C., Brevers, D., & Noël, X. (2012). On the predictive validity of automatically activated approach/avoidance tendencies in abstaining alcohol-dependent patients. Drug and Alcohol Dependence. https://doi.org/10.1016/j.drugalcdep.2012.06.019 Wiers, R. W., Boffo, M., & Field, M. (2018). What’s in a Trial? On the Importance of Distinguishing Between Experimental Lab Studies and Randomized Controlled Trials: The Case of Cognitive Bias Modification and Alcohol Use Disorders. Journal of Studies on Alcohol and Drugs, 79(3), 333–343.

Author Response

We thank you Reviewer 1 for peer reviewing our paper and for your recommendations/kind comments. Please find as enclosed our in-line replies to your recommendations. We have restructured the paper significantly and highlighted the main contribution of this paper to the academic literature.

Specific comments

  1. I don't think it does the field any favours to continually refer to approach and attentional biases as being "automatic" and "unconscious". I don't think anyone has actually provided any evidence to suggest that they are uncontrollable and unconscious. I would encourage the authors to use the phrase "relatively automatic" and remove the word 'unconscious' from the manuscript.

We agree with your recommendations and have rephrased that to “relatively automatic” and removed the terminology – that of unconscious processes. The amends are as follows:

“Attentional biases refer to the relatively automatic tendencies for attention to be preferentially allocated towards substance-related cues (1, 2). Approach biases are the relatively automatic behavioural tendencies of individuals to reach out to substance-related cues in their natural environment (2, 3).”

  1. I think the nice point made at line 31 still stands but perhaps the authors could say "conventional therapies to do not impact these more reflexive and relatively automatic processes; this is because conventional therapies target goal-directed control processes" rather than using the unconscious/conscious dichotomy.

Thank you for your recommendations. We have made the necessary changes. The amends are as follows: “The advances in experimental psychology have demonstrated that conventional psychotherapies do not impact on these more reflexive and relatively automatic processes; this is because conventional psychotherapies target conscious, goal directed cognitive control processes.”

  1. Line 58 (and 168-175; 214-218): relevant to the issue of timings of attentional bias - the authors could point out that very fast stimulus intervals (i.e. <250ms) the paradigm is more likely to capture fast/reflexive attentional orienting to signals of drug reward but that at longer stimulus intervals participants are strategically choosing where to look on screen (Godijn & Theeuwes, 2002). In the latter case this creates a lot of noise in the data and 'relatively automatic' attentional biases are no longer being measured.

We have included the suggested reference and expanded on the explanation to ensure that our readers have a better understanding of this. The amends are as follow, “Presenting stimulus images at relatively fast stimulus intervals, such as those less than 250 milliseconds, is more likely to capture the reflexive attentional orienting; as compared to presentation at longer stimulus intervals, whereby participants chose where to look on the screen. This is evident, as Garland et al. (2013) (11), have reported that in opioid users a stimulus timing interval of 200 milliseconds is better capturing biases than a 2000 millisecond interval.”

  1. Somewhere (maybe at line 70) I think the authors should point out that cognitive bias modification seems to work best for individuals who have clinical levels of dependence and are treatment seeking, rather than those who wish to merely "cut down" (Wiers et al., 2018).

Thank you for your recommendation. We have included this statement, as follows: “It is apparent that cognitive bias modification appears to work best for individuals who have clinical levels of dependence; or whom are treatment seeking (Wiers et al., 2018).”

  1. At line 72 the authors refer to a study on chronic pain in children to make the point that sometime there is no baseline bias. I think this reference seems out of place. There are plenty of studies demonstrating that there is often a lack of baseline bias in clinically dependent populations e.g.,Barkby et al., 2012; Spruyt et al., 2012) which would seem more relevant here.

Thank you for your recommended references. We have removed the reference by Sharpe et al. (2018) and added on your recommended references. We have restructured the paragraph significantly as well. The amends are as follows:

“Negative findings have been reported in studies of cognitive bias modification, and the challenges discussed above could impact on both bias assessment and modification, resulting in negative findings. Studies such as that of Barkby et al. (2012), which examined 63 individuals with alcohol dependence, reported that there was no difference amongst controls and those who received the bias modification task, in terms of approach biases. Similarly, Spruty et al. (2012) also reported the association between relapse rate and the magnitude of avoidance tendencies towards alcohol stimulus.”

  1. Related to baseline biases the authors should point out somewhere that Eberl and colleagues reported that "older patients and patients with a strong approach-bias at pretest profited most from CBM" (Eberl et al., 2013)

Thank you for the recommendations. We have included this. The amends are as follows, “Apart from negative findings, there have been documented evidence from our recent feasibility and acceptability study, in which 53% of the participants were found not to have baseline biases (15). Other studies have highlighted that the magnitude of baseline biases, in elderly patients, predicted whether bias modification was effective (Eberl C et al., 2012).”

  1. I find line 75 a bit confusing "the preparation of this perspective paper" - I was thinking - "which paper - The Sharpe one?" To avoid confusion the authors could say "This issue of baseline biases prompted the current perceptive paper after a recent feasibility and acceptability study reported.." I would make this a separate paragraph at the end of the introduction, clearly explaining what the current perspective paper is about and defining the scope.

We have introduced the main aims and scope of this perspective article in a new paragraph. It is the aim of this perspective article in scoping out each of the challenges and to discuss potential methods to mitigate against these challenges.

  1. Similarly when the authors start discussing the search of the published literature at line 84 it is unclear if they are talking about their own search or Dennis-Tiwary et al's search. This information about individual differences should surely go earlier in the introduction, before the authors start discussing their methods? I think the authors should consider have a brief 'method' section where they describe the search they did using Pubmed and MEDLINE - how many papers were returned? How did the authors decide which papers to include? If the authors do not wish to follow guidelines on systematic review then maybe it is easier to just omit the search methodology altogether (and not include a methods section). But i still think the scope of this review needs to be defined somewhere.

We have refined this. It is not our intent to conduct a systematic review and we have not followed the methods of a systematic review. We have included a statement to clarify this “A brief review of the literature was conducted in order to identify some of the relevant articles. “

  1. Line 79: I think somewhere in the introduction, when 'individual differences' are discussed the authors should give examples of what types of factors they mean.

We have done so. The amends are as follows, “Such individual differences (severity of dependence, frequency of drug use) in baseline attentional biases will impact on the effectiveness of cognitive bias modification interventions.”

Reviewer 2 Report

This perspective focuses on a summary of the attentional and approach biases researchers come across when conducting research with individuals using legal and/or illegal substances.

Although interesting as a summary of recent studies and meta-analyses, the significance of this perspective, considering very recent relevant meta-analyses and reviews from the same and other research groups, is is somehow masked. It is advisable that the authors clearly state why this perspective is important and what it offers to the already existing knowledge and publications.

Except for the above substantial consideration, there are only further minor comments and suggestions for improvement. Please find a list of them below:

  1. The abstract appears very vague. Examples of bias or challenges might help. Similarly providing examples throughout the main text, would also be very helpful to the readers.
  2. Abstract, lines 20-22, "Negative findings ... negative findings": this sentence is unclear, please rewrite. Please also make sure to remove "above" in this sentence; it seems to be copied from a section in the main text, and there is no other content above the abstract.
  3. Introduction, lines 34-35, "for the existence of these": of which?
  4. Introduction, line 57: please correct space before the citation and duplication of brackets. The same appears in other sections of the main text. Thus, please aim to proofread the perspective carefully before your next submission. E.g., line 115, line 231
  5. line 162: "variations"
  6. line 222: "is needed"
  7. line 225 "serious games": what do the authors refer to by serious games?  

Author Response

We thank you for peer reviewing our manuscript. The intents of our perspective article were in highlighting the main challenges that researchers might face when implementing cognitive bias modification interventions. This perspective article draws together the evidence for each of the challenges that we have articulated. This paper will help academics and researchers in considering the design of their future cognitive bias modification intervention.

Except for the above substantial consideration, there are only further minor comments and suggestions for improvement. Please find a list of them below:

  1. The abstract appears very vague. Examples of bias or challenges might help. Similarly providing examples throughout the main text, would also be very helpful to the readers.

We have provided example of biases – that of attentional biases and approach biases.

  1. Abstract, lines 20-22, "Negative findings ... negative findings": this sentence is unclear, please rewrite. Please also make sure to remove "above" in this sentence; it seems to be copied from a section in the main text, and there is no other content above the abstract.

We have amended and clarified this, “Negative findings, that of the absence of biases at baseline; or the lack of effectiveness of bias modification have been reported in studies of cognitive bias modification.”

  1. Introduction, lines 34-35, "for the existence of these": of which?

We have clarified that it refers to the existence of biases.

  1. Introduction, line 57: please correct space before the citation and duplication of brackets. The same appears in other sections of the main text. Thus, please aim to proofread the perspective carefully before your next submission. E.g., line 115, line 231

We have corrected this.

  1. line 162: "variations" We have corrected
  2. line 222: "is needed" We have corrected
  3. line 225 "serious games": what do the authors refer to by serious games?  We have appended the brief definition of serious game – they are games that are purposefully designed for specific interventions.

Reviewer 3 Report

In this manuscript, the authors review recent studies on cognitive (attention and approach) biases in substance use disorders and propose several challenges facing cognitive bias modification interventions. The topic is interesting and clinically important, however, the manuscript in its current form is very poorly written. The authors also keep citing their own papers throughout the manuscript without providing sufficient explanation.

1, In general, the writing is confusing and lack of clarity, the introduction of the reviewed studies is superficial and not comprehensible to a broad readership:

1), In the first sentence, the authors mention attentional and approach biases in general and in the second sentence, they suddenly define these biases in terms of substance-use disorder. Attentional and approach biases occur everywhere, not just in substance-use disorder, right? The Abstract has the same problem.

2), It seems the authors consider a) automatic, unconscious processes, b) attentional and approach biases, and c) cognitive biases as the same thing, is that accurate?

3), lines 32-41: the authors mention "several reviews that have provided evidence for the existence of these" in which "these" seem to refer to "conventional psychotherapies do not impact on these unconscious processes; this is because conventional psychotherapies target conscious, cognitive control processes". How? I cannot see any connections between the four reviews the authors introduce here with their argument. Please confirm the logic here.

4), The first challenge the authors propose is "cognitive biases are not universally present", however, I cannot find a single piece of evidence in support of this proposal. All the reviewed studies seem to have reported greater biases in more severe users (or so-called dose-dependent relationship), did any of the studies find no biases in substance users (or comparable biases to healthy subjects)? If so, please do clarify.

5), line 120-121: what do the authors mean by "correlated with the number of errors made during the testing process, and hence affected the magnitude of attentional biases"? How? The same with lines 141-142. The authors have to provide more explanation regarding the task to let their readers understand what they are talking about.

6), Line 161, what is "the visual probe paradigms"? Many of the readers are not familiar with these terms. As this relates to the second challenge, please explain the task paradigm and the "direct and indirect measures" (line 157) in detail, not just "as aforementioned in the introduction" (the introduction just had one line describing this).

7), lines 181-186: was reference 33 about anxiety or not? It is hard to infer from the current writing.

2, lines 84-86: methods of the literature search using PUBMED are somewhat unclear. The authors may provide more information regarding when and how they conducted the search.

3, The authors say "Negative findings have been reported in studies of cognitive bias modification" in the Abstract (line 20-21), however, in the main text they did not provide any in-depth discussion on studies using cognitive bias modification except citing their own papers.

Author Response

We thank you Reviewer 3 for peer reviewing our manuscript. Please find as enclosed our in-line responses to your comments.

We agree that there are other disorders with attention and approach biases. However, we have decided to define it in the context of substance use disorder, as it is the main disorder we are evaluating. We have contextualized this:

“For substance use disorders, attentional biases refer to the relatively automatic tendencies for attention to be preferentially allocated towards substance-related cues (1, 2). Approach biases are the relatively automatic behavioral tendencies of individuals to reach out to substance-related cues in their natural environment (2, 3).”

2), It seems the authors consider a) automatic, unconscious processes, b) attentional and approach biases, and c) cognitive biases as the same thing, is that accurate?

Please refer to our definition of attentional and approach biases:

“For substance use disorders, attentional biases refer to the relatively automatic tendencies for attention to be preferentially allocated towards substance-related cues (1, 2). Approach biases are the relatively automatic behavioural tendencies of individuals to reach out to substance-related cues in their natural environment (2, 3).”

3), lines 32-41: the authors mention "several reviews that have provided evidence for the existence of these" in which "these" seem to refer to "conventional psychotherapies do not impact on these unconscious processes; this is because conventional psychotherapies target conscious, cognitive control processes". How? I cannot see any connections between the four reviews the authors introduce here with their argument. Please confirm the logic here.

We like to clarify that we intent to mention that several reviews have provided evidence for cognitive biases, such as that of attentional and approach biases. Please refer to the amended paragraph:

“There are several reviews that have provided evidence for the existence of biases. For example, MacLean RR et al. (2018) (4), in their review, describe the evidence for the existence of robust attentional biases amongst individuals who were using opioids. Similarly, O’Neil et al. (2020) (5) reported that attentional biases are greater in individuals abusing cannabis. Two other reviews have focused on the effectiveness of bias modification. “

4), The first challenge the authors propose is "cognitive biases are not universally present", however, I cannot find a single piece of evidence in support of this proposal. All the reviewed studies seem to have reported greater biases in more severe users (or so-called dose-dependent relationship), did any of the studies find no biases in substance users (or comparable biases to healthy subjects)? If so, please do clarify.

We clarified that we intend to refer to individual factors and individual differences and how they affect the magnitude of attentional biases. The amends are as follow: “One of these is that the magnitude of attentional and approach biases is dependent on individual factors.” We have elaborated on individual differences:

“Such individual differences (severity of dependence, frequency of drug use) in baseline attentional biases will impact on the effectiveness of cognitive bias modification interventions. It is apparent that cognitive bias modification appears to work best for individuals who have clinical levels of dependence; or whom are treatment seeking (Wiers et al., 2018). In fact, Dennis-Tiwary et al. (2016) (16) were amongst the first to highlight how individuals differ; and that these individual differences could potentially have an impact on the effectiveness of a bias modification intervention. They highlighted that an understanding of individual differences would enable personalization of the cognitive bias modification intervention, resulting in greater modification of attentional biases.”

5), line 120-121: what do the authors mean by "correlated with the number of errors made during the testing process, and hence affected the magnitude of attentional biases"? How? The same with lines 141-142. The authors have to provide more explanation regarding the task to let their readers understand what they are talking about.

6), Line 161, what is "the visual probe paradigms"? Many of the readers are not familiar with these terms. As this relates to the second challenge, please explain the task paradigm and the "direct and indirect measures" (line 157) in detail, not just "as aforementioned in the introduction" (the introduction just had one line describing this).

We have appended more information about the nature of the assessment task. We have also rewritten the results, such that it is easier for readers to understand.

The amends are:

“The visual probe task, which has been commonly used, involves having the individual respond to the position of a probe replacing either the substance or neutral stimuli. Individuals with biases tend to respond faster to probes that replace the substance stimuli, as compared to the neutral stimulus.”

7), lines 181-186: was reference 33 about anxiety or not? It is hard to infer from the current writing.

We have removed this reference in order to avoid confusion.

2, lines 84-86: methods of the literature search using PUBMED are somewhat unclear. The authors may provide more information regarding when and how they conducted the search.

We do not intend to perform a systematic review. We have stated that we have merely provided a brief overview of the literature.

3, The authors say "Negative findings have been reported in studies of cognitive bias modification" in the Abstract (line 20-21), however, in the main text they did not provide any in-depth discussion on studies using cognitive bias modification except citing their own papers.

We have appended more references (as suggested by Reviewer 1) in order to justify the presence of negative findings.

“Negative findings have been reported in studies of cognitive bias modification, and the challenges discussed above could impact on both bias assessment and modification, resulting in negative findings. Studies such as that of Barkby et al. (2012), which examined 63 individuals with alcohol dependence, reported that there was no difference amongst controls and those who received the bias modification task, in terms of approach biases. Similarly, Spruty et al. (2012) also reported the association between relapse rate and the magnitude of avoidance tendencies towards alcohol stimulus. Apart from negative findings, there have been documented evidence from our recent feasibility and acceptability study, in which 53% of the participants were found not to have baseline biases (15). Other studies have highlighted that the magnitude of baseline biases, in elderly patients, predicted whether bias modification was effective (Eberl C et al., 2012).

Round 2

Reviewer 3 Report

Thank the authors for making the revisions.

For 1, 1) & 2), please accordingly revise the Abstract.

3), so here the authors are talking about the "evidence for the existence of biases", however, in lines 91-93, they are citing papers showing the lack of evidence (i.e., negative findings). Please reframe and make your arguments consistent and coherent. Please also refer to my comment in 3 below.

3, "Studies such as that of Barkby et al. (2012), which examined 63 individuals with alcohol dependence, reported that there was no difference amongst controls and those who received the bias modification task, in terms of approach biases." Grammar issues. This study also has nothing to do with cognitive bias modification, it just used a stimulus-response compatibility task to measure approach and avoidance biases.

"Similarly, Spruty et al. (2012) also reported the association between relapse rate and the magnitude of avoidance tendencies towards alcohol stimulus." What do the authors mean by "similarly"? Please be reminded that this is the first time the authors mention "avoidance tendencies" and Spruty et al. (2012) did not use any cognitive bias modification. Please do read Barkby et al. (2012) and Spruty et al. (2012) carefully and revise this part. Both Barkby et al. (2012) and Spruty et al. (2012) are missing in References.

Since neither Barkby et al. (2012) nor Spruty et al. (2012) is relevant to cognitive bias modification, the statement "Negative findings have been reported in studies of cognitive bias modification" should be accordingly revised or supported with other evidence.

Author Response

We thank you Peer Reviewer 3 for reviewing our manuscript again, and for your suggested changes. We have taken them into consideration and amended the manuscript accordingly. Please find as enclosed our in-line replies to your comments.

For 1, 1) & 2), please accordingly revise the Abstract.

We have amended the abstract accordingly. The amended abstract is as follows:

Abstract: In recent years, the advances in experimental psychology have led to there being better understanding of automatic, unconscious processes, referred to as attentional and approach biases amongst individuals with substance use disorders. Attentional biases refer to the relatively automatic tendencies for attention to be preferentially allocated towards substance-related cues; whereas approach bias is the relatively automatic behavioral tendencies of individuals to reach out to substance-related cues in their natural environment. Whilst several reviews confirm the existence of these biases, and the effectiveness of bias modification, the conduct of cognitive bias modification amongst substance-using individuals is not without its challenges. One of these is that cognitive biases, both attentional and approach biases, are not universally present; and several individual differences factors modulate the magnitude of the biases. Another challenge that investigators faced in their conduct of cognitive bias modification relates to the selection of the appropriate task for bias assessment and modification. Other challenges intrinsic to cognitive bias modification intervention relates to that of participant attrition, much like conventional psychotherapies. Negative findings, that of the absence of biases at baseline; or the lack of effectiveness of bias modification have been reported in studies of cognitive bias modification. All these challenges could have an impact on bias assessment and modification. In this perspective paper, we will explore the literature surrounding each of these challenges and discuss potential measures that could be undertaken to mitigate these clinical and research challenges.”

We have defined both attentional and approach biases and stated that these biases are found in substance using individuals. We have clarified that the reviews done to date demonstrate the existence of biases and the effectiveness of bias modification amongst substance-using individuals. We hope these amends suffice and render the abstract more readable and comprehensible by the intended readers.

3), so here the authors are talking about the "evidence for the existence of biases", however, in lines 91-93, they are citing papers showing the lack of evidence (i.e., negative findings). Please reframe and make your arguments consistent and coherent. Please also refer to my comment in 3 below.

We thank you for your comments. Both these references were considered, as they were recommended previously by Reviewer 1, and hence they were included. We have reconsidered both these references, and critically appraised the references/articles again – indeed they are not appropriate.

Please find the amends made to the paragraph. We have restructured the paragraph in order for the arguments to be more consistent; and we have amended the references to introduce more references detailing negative findings.

“One last challenge that researchers face with regards to bias modification interventions pertains to that of negative findings. Negative findings could refer to either the absence of baseline biases amongst individuals during the assessment stage; or the absence of positive findings following bias modification. Our previous study, which was a feasibility and acceptability study examining a mobile attention bias modification intervention revealed that 53% of the participants sampled to have no baseline biases (16). Dean AC et al. (2019) (17) in their recently concluded study examining attentional bias modification amongst 42 methamphetamine-dependent individuals reported negative findings following intervention, i.e. bias retraining did not reduce attentional biases. Zhu Y et al. (2018) (18) in their prior study examining a computerized cognitive addiction therapy application for individuals with methamphetamine disorders also reported that the cognitive bias retraining functionality in the app did not reduce overall biases. There have also been other studies have highlighted that the magnitude of baseline biases, in elderly patients, predicted whether bias modification was effective (Eberl C et al., 2012) (19). There warrants further understanding of why there are negative findings, and how baseline biases correlate with eventual outcomes of bias retraining.”

3, "Studies such as that of Barkby et al. (2012), which examined 63 individuals with alcohol dependence, reported that there was no difference amongst controls and those who received the bias modification task, in terms of approach biases." Grammar issues. This study also has nothing to do with cognitive bias modification, it just used a stimulus-response compatibility task to measure approach and avoidance biases.

"Similarly, Spruty et al. (2012) also reported the association between relapse rate and the magnitude of avoidance tendencies towards alcohol stimulus." What do the authors mean by "similarly"? Please be reminded that this is the first time the authors mention "avoidance tendencies" and Spruty et al. (2012) did not use any cognitive bias modification. Please do read Barkby et al. (2012) and Spruty et al. (2012) carefully and revise this part. Both Barkby et al. (2012) and Spruty et al. (2012) are missing in References.

Since neither Barkby et al. (2012) nor Spruty et al. (2012) is relevant to cognitive bias modification, the statement "Negative findings have been reported in studies of cognitive bias modification" should be accordingly revised or supported with other evidence.

We apologized for the mix-up in the references. We have included all the additional references/articles.

References

  1. Field M, Marhe R, Franken IH. The clinical relevance of attentional bias in substance use disorders. CNS Spectr. 2014;19(3):225-30.
  2. Cristea IA, Kok RN, Cuijpers P. The Effectiveness of Cognitive Bias Modification Interventions for Substance Addictions: A Meta-Analysis. PLoS One. 2016;11(9):e0162226.
  3. Manning V, Mroz K, Garfield JBB, Staiger PK, Hall K, Lubman DI, et al. Combining approach bias modification with working memory training during inpatient alcohol withdrawal: an open-label pilot trial of feasibility and acceptability. Substance abuse treatment, prevention, and policy. 2019;14(1):24.
  4. MacLean RR, Sofuoglu M, Brede E, Robinson C, Waters AJ. Attentional bias in opioid users: A systematic review and meta-analysis. Drug Alcohol Depend. 2018;191:270-8.
  5. O'Neill A, Bachi B, Bhattacharyya S. Attentional bias towards cannabis cues in cannabis users: A systematic review and meta-analysis. Drug Alcohol Depend. 2019:107719.
  6. Boffo M, Zerhouni O, Gronau QF, et al. Cognitive Bias Modification for Behavior Change in Alcohol and Smoking Addiction: Bayesian Meta-Analysis of Individual Participant Data. Neuropsychol Rev. 2019;29(1):52-78. doi:10.1007/s11065-018-9386-4
  7. Bearre L, Sturt P, Bruce G, Jones BT. Heroin-related attentional bias and monthly frequency of heroin use are positively associated in attenders of a harm reduction service. Addict Behav. 2007;32(4):784-92.
  8. Wiers, R. W., Boffo, M., & Field, M. (2018). What’s in a Trial? On the Importance of Distinguishing Between Experimental Lab Studies and Randomized Controlled Trials: The Case of Cognitive Bias Modification and Alcohol Use Disorders. Journal of Studies on Alcohol and Drugs79(3), 333–343.
  9. Dennis-Tiwary TA, Egan LJ, Babkirk S, Denefrio S. For whom the bell tolls: Neurocognitive individual differences in the acute stress-reduction effects of an attention bias modification game for anxiety. Behav Res Ther. 2016;77:105-17.
  10. Zhang MW, Ying JB, Song G, Fung DSS, Smith HE. Recent Advances in Attention Bias Modification for Substance Addictions. Int J Environ Res Public Health. 2018;15(4).
  11. Zhang M, Fung DSS, Smith H. Variations in the Visual Probe Paradigms for Attention Bias Modification for Substance Use Disorders. Int J Environ Res Public Health. 2019;16(18).
  12. Godijn, R., & Theeuwes, J. (2002). Programming of endogenous and exogenous saccades: Evidence for a competitive integration model. Journal of Experimental Psychology. Human Perception and Performance28(5), 1039–1054Garland EL, Froeliger BE, Passik SD, Howard MO. Attentional bias for prescription opioid cues among opioid dependent chronic pain patients. J Behav Med. 2013;36(6):611-20.
  13. Zhang M, Ying J, Song G, Fung DS, Smith H. Gamified Cognitive Bias Modification Interventions for Psychiatric Disorders: Review. JMIR Ment Health. 2018;5(4):e11640.
  14. Zhang M, Ying J, Song G, Fung DS, Smith H. Attention and Cognitive Bias Modification Apps: Review of the Literature and of Commercially Available Apps. JMIR Mhealth Uhealth. 2018;6(5):e10034.
  15. Zhang M, Ying J, Amron SB, Mahreen Z, Song G, Fung DSS, et al. A Smartphone Attention Bias App for Individuals With Addictive Disorders: Feasibility and Acceptability Study. JMIR Mhealth Uhealth. 2019;7(9):e15465.
  16. Dean AC, Nurmi EL, Moeller SJ, Amir N, Rozenman M, Ghahremani DG, et al. No effect of attentional bias modification training in methamphetamine users receiving residential treatment. Psychopharmacology (Berl). 2019;236(2):709-21.
  17. Zhu Y, Jiang H, Su H, Zhong N, Li R, Li X, et al. A Newly Designed Mobile-Based Computerized Cognitive Addiction Therapy App for the Improvement of Cognition Impairments and Risk Decision Making in Methamphetamine Use Disorder: Randomized Controlled Trial. JMIR Mhealth Uhealth. 2018;6(6):e10292.
  18. Eberl, C., Wiers, R. W., Pawelczack, S., Rinck, M., Becker, E. S., & Lindenmeyer, J. (2013). Approach bias modification in alcohol dependence: Do clinical effects replicate and for whom does it work best? Developmental Cognitive Neuroscience4, 38–51. https://doi.org/10.1016/j.dcn.2012.11.002 
  19. Fadardi JS, Ziaee SS. A comparative study of drug-related attentional bias: evidence from Iran. Exp Clin Psychopharmacol. 2010;18(6):539-45.
  20. Constantinou N, Morgan CJ, Battistella S, O'Ryan D, Davis P, Curran HV. Attentional bias, inhibitory control and acute stress in current and former opiate addicts. Drug Alcohol Depend. 2010;109(1-3):220-5.
  21. Anderson BA, Faulkner ML, Rilee JJ, Yantis S, Marvel CL. Attentional bias for nondrug reward is magnified in addiction. Exp Clin Psychopharmacol. 2013;21(6):499-506.
  22. Waters AJ, Marhe R, Franken IH. Attentional bias to drug cues is elevated before and during temptations to use heroin and cocaine. Psychopharmacology (Berl). 2012;219(3):909-21.
  23. Field M, Eastwood B, Bradley BP, Mogg K. Selective processing of cannabis cues in regular cannabis users. Drug Alcohol Depend. 2006;85(1):75-82.
  24. Cousijn J, Goudriaan AE, Wiers RW. Reaching out towards cannabis: approach-bias in heavy cannabis users predicts changes in cannabis use. Addiction. 2011;106(9):1667-74.
  25. Cousijn J, Watson P, Koenders L, Vingerhoets WA, Goudriaan AE, Wiers RW. Cannabis dependence, cognitive control and attentional bias for cannabis words. Addict Behav. 2013;38(12):2825-32.
  26. Campbell DW, Stewart S, Gray CEP, Ryan CL, Fettes P, McLandress AJ, et al. Chronic cannabis use and attentional bias: Extended attentional capture to cannabis cues. Addict Behav. 2018;81:17-21.
  27. Field M. Cannabis 'dependence' and attentional bias for cannabis-related words. Behav Pharmacol. 2005;16(5-6):473-6.
  28. Vujanovic AA, Wardle MC, Liu S, Dias NR, Lane SD. Attentional bias in adults with cannabis use disorders. J Addict Dis. 2016;35(2):144-53.
  29. Marks KR, Roberts W, Stoops WW, Pike E, Fillmore MT, Rush CR. Fixation time is a sensitive measure of cocaine cue attentional bias. Addiction. 2014;109(9):1501-8.
  30. Boendermaker WJ, Sanchez Maceiras S, Boffo M, Wiers RW. Attentional Bias Modification With Serious Game Elements: Evaluating the Shots Game. JMIR Serious Games. 2016;4(2):e20.
  31. Lau HM, Smit JH, Fleming TM, Riper H. Serious Games for Mental Health: Are They Accessible, Feasible, and Effective? A Systematic Review and Meta-analysis. Front Psychiatry. 2016;7:209.
  32. Zhang M, Heng S, Song G, Fung DS, Smith HE. Co-designing a Mobile Gamified Attention Bias Modification Intervention for Substance Use Disorders: Participatory Research Study. JMIR Mhealth Uhealth. 2019;7(10):e15871.

Round 3

Reviewer 3 Report

Thank the authors for making the revisions. I think the two references recommended by Reviewer 1 are important and worth mentioning in the present manuscript. Both Barkby et al. (2012) and Spruty et al. (2012) looked at approach/avoidance biases in patients undergoing detoxification. The authors just need to read these papers carefully and include them in the current story.

Author Response

Dear Reviewer, 

We have re-examined the two papers. We have included them under negative findings, as the results arising from each of those papers do not indicate the presence of approach biases amongst the sampled participants.

The revisions are:

"

Negative findings could refer to either the absence of baseline biases amongst individuals during the assessment stage; or the absence of positive findings following bias modification. Our previous study, which was a feasibility and acceptability study examining a mobile attention bias modification intervention revealed that 53% of the participants sampled to have no baseline biases (16). Dean AC et al. (2019) (17) in their recently concluded study examining attentional bias modification amongst 42 methamphetamine-dependent individuals reported negative findings following intervention, i.e. bias retraining did not reduce attentional biases. Zhu Y et al. (2018) (18) in their prior study examining a computerized cognitive addiction therapy application for individuals with methamphetamine disorders also reported that the cognitive bias retraining functionality in the app did not reduce overall biases. For approach/avoidance biases, Barkby et al. (2012) (19) have previously examined 63 alcohol-dependent participants undergoing detoxification against 64 controls, who were light drinkers. They found, based on their stimulus-compatibility task, that there were no differences between dependent drinker and light drinkers on the task. The only exception was that those who drank more had a greater magnitude of approach biases. Spruyt A et al. (2012) (20) also examined for approach/avoidance biases amongst individuals who were alcohol dependent and amongst controls. The authors reported that their sampled participants demonstrated avoidance tendencies to stimulus, instead of the typical approach tendencies. There have also been other studies have highlighted that the magnitude of baseline biases, in elderly patients, predicted whether bias modification was effective (Eberl C et al., 2012) (21). There warrants further understanding of why there are negative findings, and how baseline biases correlate with eventual outcomes of bias retraining."